# Critical Review of Solidification of Sandy Soil by Microbially Induced Carbonate Precipitation (MICP)

Liuxia Chen [1,2], Yuqi Song [3,*], Jicheng Huang [4], Chenhuan Lai [1,2], Hui Jiao [1,2], Hao Fang [5], Junjun Zhu [1,2] and Xiangyang Song [1,2,*]

1   College of Chemical Engineering, Nanjing Forestry University, Nanjing 210037, China; clx@njfu.edu.cn (L.C.); LCH2014@njfu.edu.cn (C.L.); jiaohui2021@outlook.com (H.J.); zhujj@njfu.edu.cn (J.Z.)
2   Key Laboratory of Forestry Genetics & Biotechnology, Ministry of Education, Nanjing Forestry University, Nanjing 210037, China
3   Department of Civil Engineering, Monash University, Melbourne 3800, Australia
4   Department of Engineering Mechanics, School of Civil Engineering, Southeast University, Nanjing 210096, China; 220190985@seu.edu.cn
5   College of Life Sciences, Northwest A&F University, Xianyang 712100, China; fanghao@nwsuaf.edu.cn
*   Correspondence: yuqi.song@monash.edu (Y.S.); xysongnanlin@njfu.edu.cn (X.S.)

**Abstract:** Microbially induced carbonate precipitation (MICP) is a promising technology for solidifying sandy soil, ground improvement, repairing concrete cracks, and remediation of polluted land. By solidifying sand into soil capable of growing shrubs, MICP can facilitate peak and neutralization of $CO_2$ emissions because each square meter of shrub can absorb 253.1 grams of $CO_2$ per year. In this paper, based on the critical review of the microbial sources of solidified sandy soil, models used to predict the process of sand solidification and factors controlling the MICP process, current problems in microbial sand solidification are analyzed and future research directions, ideas and suggestions for the further study and application of MICP are provided. The following topics are considered worthy of study: (1) MICP methods for evenly distributing $CaCO_3$ deposit; (2) minimizing $NH_4^+$ production during MICP; (3) mixed fermentation and interaction of internal and exogenous urea-producing bacteria; (4) MICP technology for field application under harsh conditions; (5) a hybrid solidification method by combining MICP with traditional sand barrier and chemical sand consolidation; and (6) numerical model to simulate the erosion resistance of sand treated by MICP.

**Keywords:** micro-organism; urease; curing; sandy soil; desertification

## 1. Introduction

Microbially induced carbonate precipitation (MICP) is a promising technology applied to many civil and environmental engineering scenarios, especially combating desertification [1]. Desertification refers to the process of land degradation in arid, early semi-dry, and arid and subhumid areas under the action of various factors, including climate and human activities. The severe problem of global desertification is caused by natural and human factors, climate change, wind and rain erosion of the soil, the pursuit of economic benefits, destruction of vegetation, and unreasonable use of water resources, all of which aggravate the formation of desertification. Because the emergence of desertification has caused a significant impact on the environment and economical construction, it is highly urgent to control it [2]. Kimura et al. [3] classified and counted the global dry areas in 2017 according to the satellite-based aridity index (SbAI). Figure 1 [3] shows areas of global arid areas in 2017 and their proportion in the total land area. It can be seen from Figure 1 [3] that the total area of global arid areas accounts for 41% of the total land area. Figure 2 [4] shows the global desertification risk level distribution map from 2000 to 2014, estimated based on the global Desertification Vulnerability Index and the ratio of areas with different risk levels. The colors in Figure 1 represent different levels of global desertification risk.

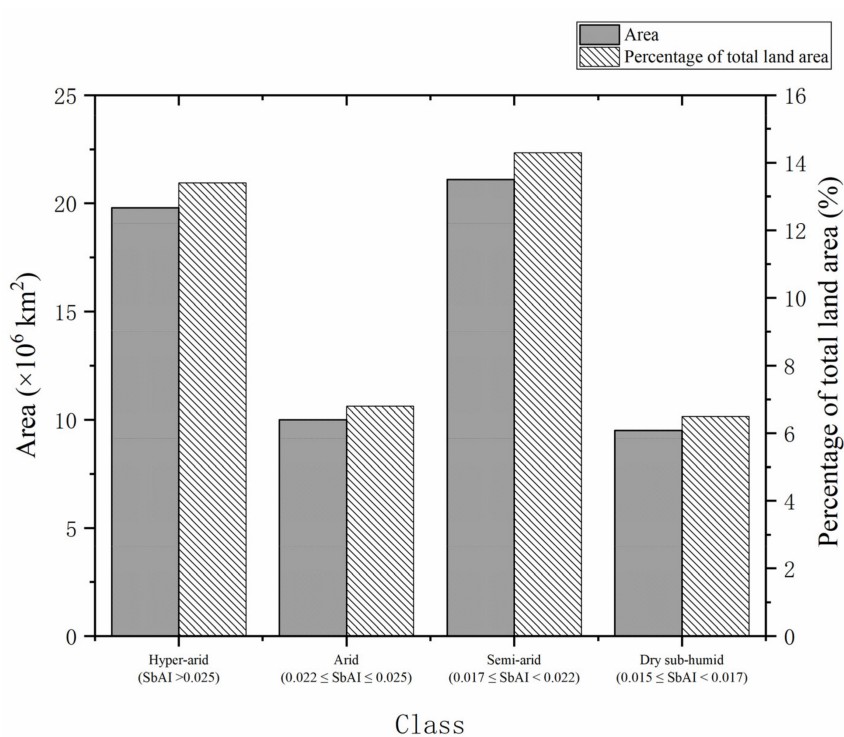

**Figure 1.** Schematic diagram of arid areas worldwide and their percentage of total land area in 2017 based on the Satellite Drought Index (SbAI) [3].

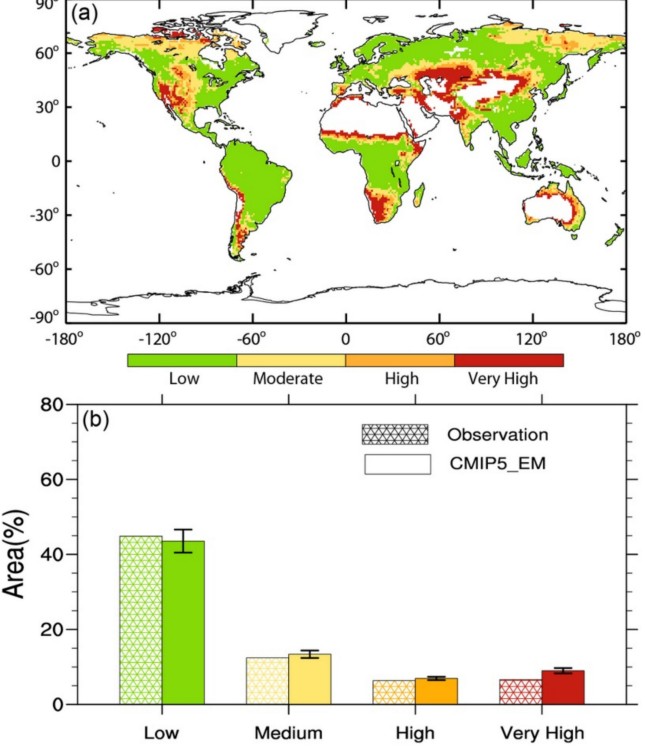

**Figure 2.** (**a**) The global desertification risk level distribution from 2000 to 2014, estimated based on the Global Desertification Vulnerability Index; (**b**) The proportion of areas with different desertification risk levels in the global land area [4].

Sand consolidation uses various ways to reduce the sand porosity and fix the sand particles [5]. As one of the common ways of consolidation, grouting consolidation is defined as

applying pressurized grouting slurry to infiltrate within the void of sandy soil, followed by the compaction and solidification of sand along with the slurry [6]. The grouting method usually includes the chemical grouting method and the biological grouting method [7,8]. In the early stage, methods of sand consolidation in desertification were limited to chemical grouting with cement, lime, and other chemical materials. However, various sand consolidation methods emerged following extensive research, including sand fixation with the sand barrier, chemicals, and microbial grouting [7–9]. The grouting method can only be used for coarse sand with a particle size greater than 4.75 mm, and microbial sand consolidation can be used for fine or medium sand with less than 0.6 mm [10]. Chemical grouting methods cost less, but chemical grouting materials (e.g., cement, lime, or adhesive) are harmful to the environment. The microbial grouting method has a relatively high cost but is friendly to the environment and can effectively improve the properties of sand [7,8]. Figure 3 [6] shows a schematic diagram of the grouting method. Microbial sand fixation refers to adding cementation solution to stimulate bacteria and then forming calcium carbonate crystals in the sand to consolidate the sand. Cementation solution generally refers to a mixture of calcium salts, nutrients and urea. Figure 4 [11] shows a diagram of the experimental setup for MICP. Microbial sand fixation not only has the advantages of environmental protection, low pollution, effective maintenance of soil moisture in sandy deserts, improvement of soil fertility, and improvement of soil thermal conductivity, but can also turn the sandy desert into soil and increase the area of state-owned arable land, which is of practical significance for the curbing of desertification. Countries around the world put forward the strategic goal of carbon peak, and carbon neutrality due to global climate change leads to many extreme climate events. "Peak carbon" refers to when carbon dioxide emissions reach a peak and then stop rising and gradually fall back. Carbon neutrality means achieving zero carbon dioxide emissions by offsetting total greenhouse gas emissions through afforestation, energy conservation, and emission reduction. The United States and the European Union have announced carbon neutrality by 2050 and China by 2060. Microbial sand fixation can also facilitate carbon peaking and carbon neutralization because microbial sand solidification technology can enable the sand to grow shrubs. Li et al. [12] found that each square meter of shrub can absorb 253.1 grams of carbon dioxide per year. All in all, microbial sand fixation has gradually become an important research topic [13,14].

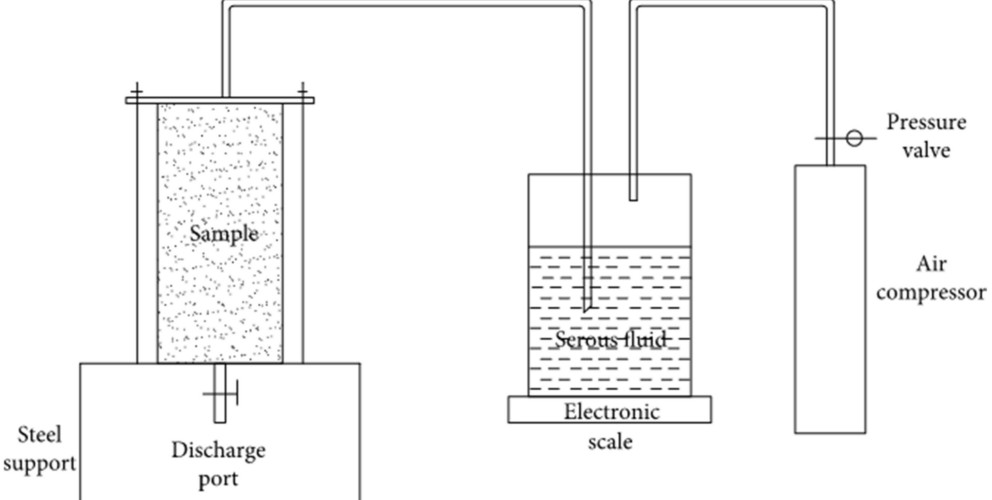

**Figure 3.** Schematic diagram of grouting consolidation device [6].

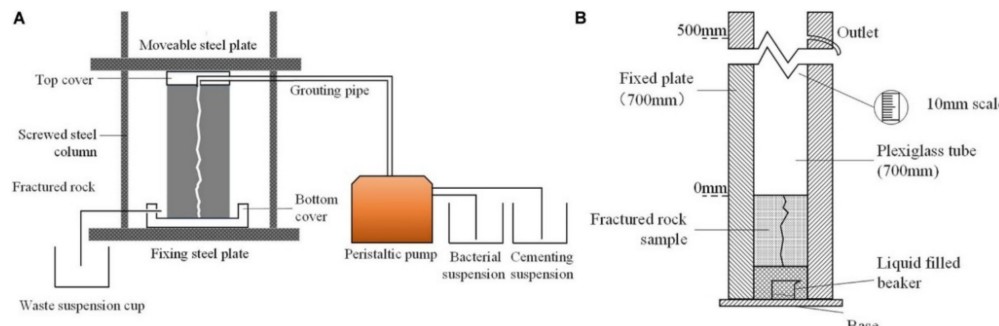

**Figure 4.** Schematic diagram of MICP experimental device. (**A**) Grouting device; (**B**) Seepage device [11].

The source of the urease-producing microbe of MICP can be indigenous or exogenous [15–17]. If the urease activity of the local (indigenous) urease-producing bacteria is high, the local bacteria can be directly used for the sand fixation. However, if the activity of local urease-producing bacteria is low, the cementation solution with nutrients should be added to stimulate the local bacteria to produce enough urease. In addition, exogenous urease-producing bacteria can be added to increase the effect of microbial sand consolidation. After treatment with MICP technology, the porosity and water conductivity of soil are reduced due to the combination of $CaCO_3$ precipitation and the medium, and the soil after solidification is not easily liquified by the action of earthquakes [18–20]. Due to the tiny pores in the clays, it is difficult for the bacteria to enter the clays, and therefore there are few studies on the microbial solidification of clays. Liu et al. [21] studied the effect of MICP on the repair of dry cracks in clays. Sun et al. [22] treated a sand-clay mixture with MICP and found that different amounts of clays need to be added to solidify sand with different particle sizes.

Microbial sand fixation technology is not limited to small-scale laboratory studies but is also applied to large-scale outdoor studies. Figure 5 [23] and Figure 6 [24] are photos of extensive outdoor experiments. Figure 7 [25] is a schematic diagram of the test system. Meng et al. [26] conducted outdoor experiments in Ulan Buh Desert, and the results confirmed that the use of *Sporosarcina pasteurii* to consolidate desert soil could improve the wind erosion resistance of soil. Outdoor microbial sand fixation experiments need to overcome many difficulties and the cost is relatively high. According to the results of small sand fixation experiments in the laboratory, part of the results of large outdoor experiments can be predicted by using the model [27].

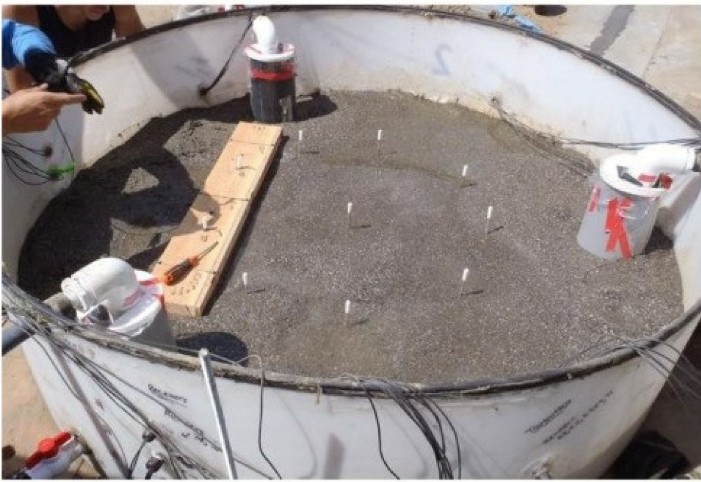

**Figure 5.** Photos of microbial sand fixation in the field [23].

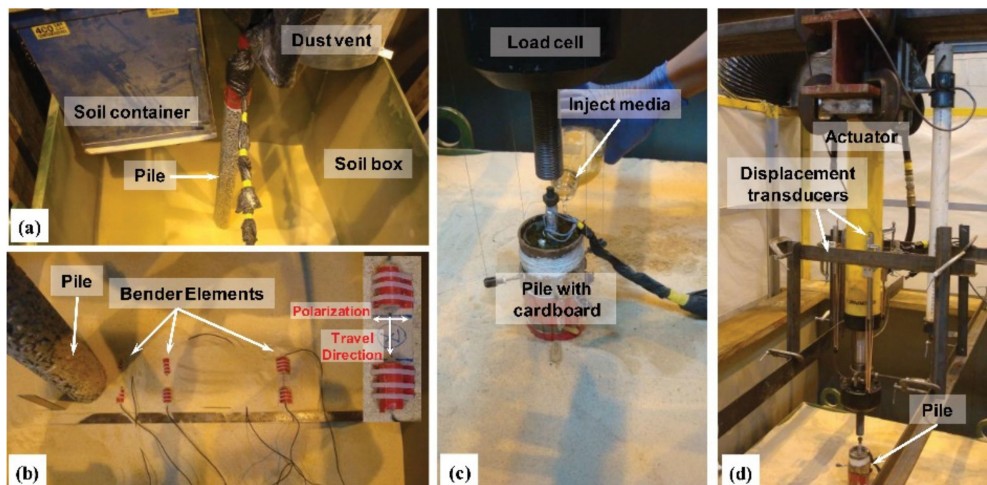

**Figure 6.** Detailed device drawing for large MICP grouting (**a**) soil raining; (**b**) bender element installation; (**c**) media injected from the top of the pile; (**d**) pull-out loading setup [24].

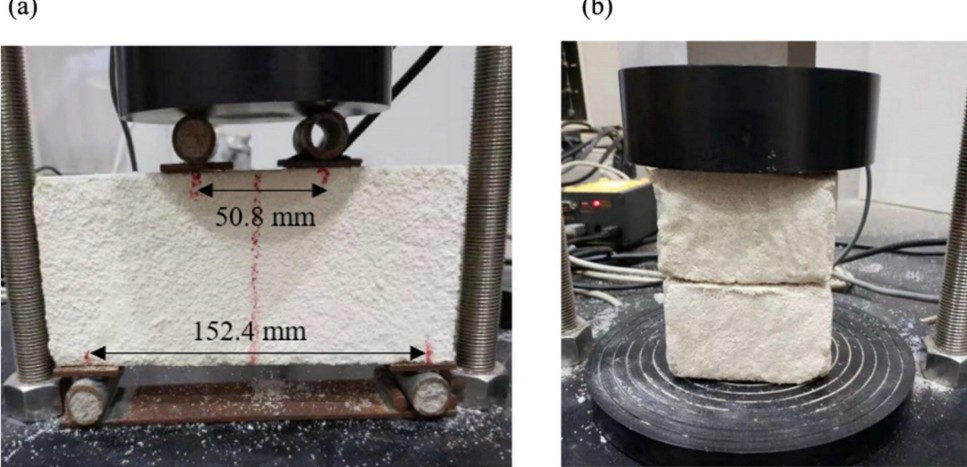

**Figure 7.** Schematic diagram of MICP test system. (**a**) Four-point bending test system; (**b**) Compressive strength testing system [25].

Microbial sand consolidation is affected by many factors, such as the concentration of the cementation solution, the concentration of culture liquid, temperature, calcium source and pH value. After optimizing various factors, the solidification effect of the sand body after microbial solidification can be effectively improved [28]. Micro-organisms can induce $CaCO_3$ to cement the curing medium, and the degree of curing needs to be measured or characterized by microscopic photographs taken by precision instruments. Scanning electron microscopy (SEM), light microscopy and other instruments are usually used to observe the generated calcium carbonate crystals, which can more intuitively reflect the effect of microbial sand consolidation. There are other methods to characterize the solidification effect of microbial cementation based on microscopic photographs, the determination of $CaCO_3$ content, permeability, shear wave velocity, Fourier transforms infrared spectrum analysis, and scanning electron microscopy [29–31].

MICP technology has dramatically developed in the past ten years in laboratory-scale solidified sand. However, there are still many challenges to overcome in applying MICP to field-scale practical engineering. Due to the long study cycle of MICP sand treatment and the high cost of large-scale field operations, few studies have been conducted on large-scale outdoor solidified sand [1]. In addition, the application of MICP to the harsh environment, including high temperatures, freeze–thaw cycles, wet–dry conditions, and acid rain, needs further study [32]. Ammonium ions produced during MICP can be hazardous to the

environment if left untreated [33]. The removal method of the ammonium ion should also be further studied in the future.

In this paper, progress in the research on microbial mineralized sandy soil is summarized, including ways of solidifying sandy soil, microbial sources of solidified sand, models used to predict the curing process of MICP in the field, factors that influence microbial solidification of sandy soil, the analysis of current problems and discussion on the prospects for the application of microbial sand consolidation technology in the future.

## 2. Methods of Solidifying Sand

As desertification increases, the local ecological environment and economy are greatly affected, and the continuous expansion of areas of desertification correspondingly reduces the arable area [34]. If desertification can be effectively contained, it will be beneficial to the ecological environment and economic development. In order to curb the spread of desertification, the government and relevant departments have proposed many solutions, such as reducing the destruction of forests and protecting vegetation [35]. At present, the main methods of solidifying sand include microbial sand consolidation and other sand fixation methods. Each method has advantages and disadvantages. In practical applications, the most appropriate cure should be selected according to specific conditions, such as the nature of the solidified sample and the external environment [36].

### 2.1. Microbial Sand Consolidation

In the past, researchers focused on the geological processes of soil. With the rapid development of microbiology, researchers began to discuss the impact of microbial activity on the mechanical properties of soil. Microbial sand consolidation technology is a product of microbial application in the field of geotechnical engineering [37]. Large-scale experiments have verified that microbial sand consolidation can improve the foundation and increase the strength of granular soil through microbial grouting [38]. Venuleo et al. [39] solidified silica sand with *Sporosarcina pasteurii*; the grain density of silica sand is 2.65 g/cm$^3$, the concentrations of calcium chloride and urea were both 0.25 M, and the batch grouting times were 16. The results showed that the calcium carbonate content of the solidified sand was 0.13 g/cm$^3$ and the thermal conductivity of the soil was increased by 250% compared with that of untreated soil. The thermal conductivity of sand refers to the heat transferred through a unit horizontal cross-sectional area per unit time. The increase in thermal conductivity was due to the formation of calcium carbonate, which increased the contact area between sand particles and thus promoted heat transfer.

MICP can utilize four types of bacteria: urease-producing bacteria, denitrifying bacteria, sulfate-reducing bacteria, and ferric-reducing bacteria. It is most commonly used in the laboratory to strengthen sandy soil by urea hydrolysis of urease-producing bacteria [40]. Microbial sand consolidation by urease-producing bacteria is achieved by adding the microbial solution to sand, then adding the cementation solution consisting of urea and calcium chloride solution after standing for a while. The microorganisms used in the cultured liquid are preferably slightly aerobic or facultative anaerobic [41]. There are many species of urea-producing bacteria, including *B. megaterium*, *Pararhodobacter* sp., and *Sporosarcina pasteurii*. Table 1 lists some urea-producing bacteria which have been found and the sources of these bacteria.

**Table 1.** MICP research using various urea-producing bacteria.

| The Name of the Bacteria | The Sources of the Bacteria | References, Year |
|---|---|---|
| *Sporosarcina pasteurii* MTCC 1761 | The Institute of Microbial Technology, Chandigarh, India | [42], 2009 |
| *B. megaterium* ATCC 14581 | American Type Culture Collection | [43], 2013 |
| *Pararhodobacter* sp. | The soil near beach rock in Sumuide, Nago Okinawa, Japan | [44], 2018 |
| *Staphylococcus saprophyticus* subsp. *saprophyticus* *Sporosarcina globispora* *Bacillus lentus* strain NCIMB8773 *Sporosarcina* sp. | Calcareous sand and limestone cave soils | [45], 2016 |
| *Bacillus pasteurii* NCIM 2477 *Brevibacterium ammoniagenes* ATCC 6871 *Bacillus lentus* 2466-NCIB 8773 | The National Collection of Industrial Microorganisms (NCIM) | [46], 2009 |
| *B. licheniformis* ATCC 14580 *S. pasteurii* ATCC 11859 | American Type Culture Collection | [47], 2019 |
| *S. pasteurii* BNCC 337394 | BeNa Culture Collection | [48], 2020 |
| *Bacillus* sp. DSM 23526 | Deutsche Sammlung von Mikroorganismen und Zellkulturen (DSMZ) | [49], 2014 |
| *E. undae* YR10 | Isolated from Yangtze River near Chongming County, Shanghai, China | [50], 2014 |
| *Bacillus velezensis* | Isolated from native Indian soil | [51], 2020 |
| *Pseudomonas nitroreducens* szh_asesj15 | Isolated from landfill groundwater | [52], 2021 |
| *Bacillus* sp. xjlu_herc15 *Bacillus licheniformis* adseedstjo15 | Isolated from leachate Isolated from leachate | |
| *Lysinibacillus xylanilyticus* *Psychrobacillus* sp | Isolated from Hokkaido, Japan | [53], 2019 [54], 2019 |

The bacterium commonly used in sand consolidation is *Sporosarcina pasteurii* [55]. The MICP principle based on urease-producing bacteria is that urease-producing bacteria produce urease in a suitable growth environment, and urease promotes the hydrolysis of urea to produce $CO_3^{2-}$, as it reacts with $Ca^{2+}$ to produce $CaCO_3$ precipitation. The specific reaction equations are shown in Equations (1) and (2). It can be seen from Equation (1) that the microbial sand consolidation with urease bacteria will produce the by-product of $NH_4^+$, and the high concentration of $NH_4^+$ will harm the environment if not treated [56]. There are many studies on ammonium removal. Keykha et al. [57] used *Sporoscarcina pasteurii* to generate carbonate ions in the process of culture. Then, natural zeolite was used to remove ammonium ions generated in the process of culture order to prevent the damage of ammonium ions to the soil environment, including vegetation and groundwater resources. The experiment results showed that negatively charged zeolite could absorb $NH_4^+$ to standard levels (i.e., less than 0.5 mg/L). Gowthaman et al. [58] used struvite to significantly reduce ammonium produced as a by-product during MICP by a two-step method. In the first stage, the conditions of rinsing were studied to optimize ammonia removal from soil. In the second stage, the influence of the pH condition, ammonia mole ratio, and calcium ion on struvite precipitation were studied. Research showed that struvite precipitation could remove about 90% of ammonia. Gowthaman et al. [59] controlled the pH of the reaction process by changing the content of urea so that the curing process was in a different pH range, and the morphology of the generated calcium phosphate precipitation changed with the change of pH. The experiment results showed that calcium phosphate biocementation at pH 3.4–7.5 can reduce the release of ammonium ions by about 50% and toxic ammonia by approximately 90% in the environment in comparison with conventional biocementation [59]. Therefore, future studies can be focused on minimizing the by-product of $NH_4^+$ to make the MICP technology more environmental-friendly. Figure 8 [60] shows the schematic diagram of MICP based on urease-producing bacteria. Figure 9 [61] shows the processing steps, and detection process of MICP applied to solidified sand.

$$CO(NH_2)_2 + 3H_2O = HCO_3^- + 2NH_4^+ + OH^- \tag{1}$$

$$Ca^{2+} + HCO_3^- + OH^- = CaCO_3 \downarrow + H_2O \tag{2}$$

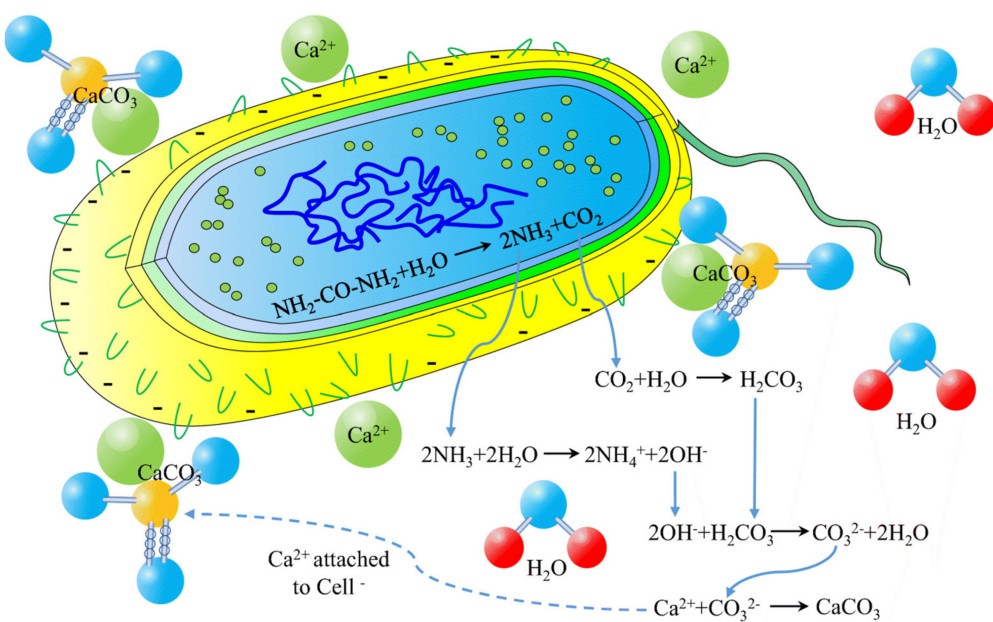

**Figure 8.** Schematic diagram of MICP based on urease-producing bacteria [60].

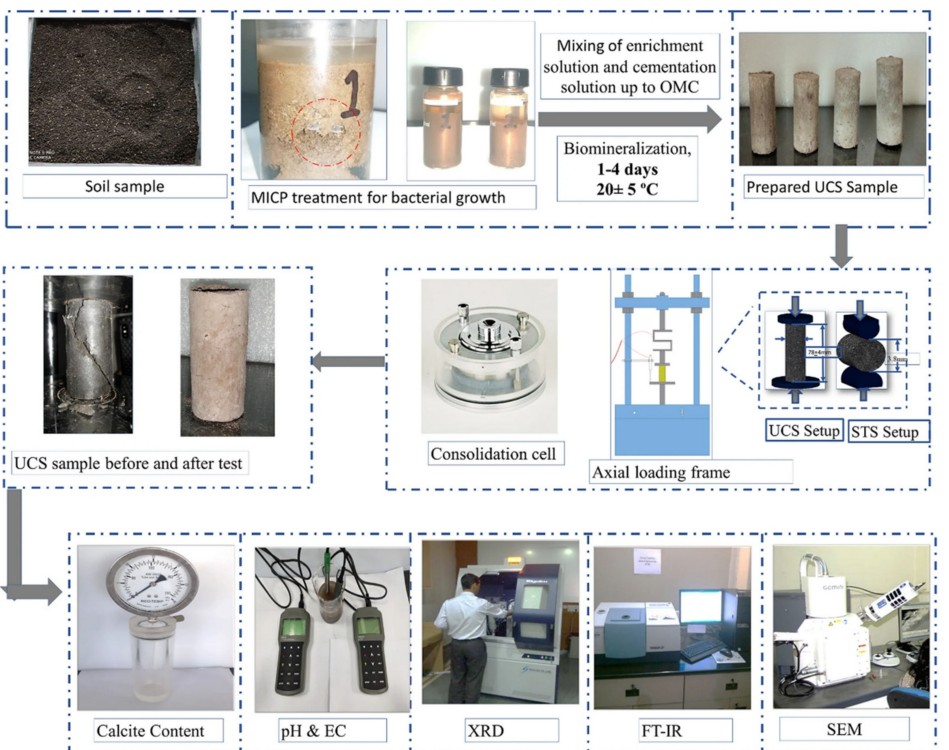

**Figure 9.** Detailed treatment procedures and testing procedures for microbial solidified sand [61].

The stability of the mechanical properties of bio-consolidated sand can also be affected by freeze–thaw cycles, wet–dry conditions, and acid rain. MICP-treated slope soil has a certain tolerance to freeze–thaw erosion, and the degree of cementation largely determines the overall stability [62]. Sharma et al. [63] studied the effects of freeze–thaw cycles on shear strength and shear modulus of Narmada river sand treated with MICP under different conditions. The experiment results showed that after 5 and 10 freeze–thaw

cycles, the strength of all biologically treated samples decreased by less than 5% and 10%. Sun et al. [32] found that slopes treated with MICP-polyacrylamide have good freeze–thaw durability and soil loss was still small after 12 freeze–thaw cycles. Gowthaman et al. [62] found that when subjected to 25 cycles of frost, samples bonded to an average of 11–13% ($CaCO_3$ mass) were high eroded (50% mass loss), while samples bonded to 20–23% only slightly eroded (2% mass loss). Sharma et al. [64] found that after 20 wet–dry cycles, the mass loss rate of samples with calcite precipitation amount (10.2–12%) was less than 3%, and the total mass of the minimum precipitation samples (3.25–3.89%) remained above 70%. Gowthaman et al. [65] studied the influence of acid rain conditions on the durability of MICP-treated slope soil and found that when $CaCO_3$ content was 12.5%, the soil loss rate was 19.9%, while when cemented $CaCO_3$ content was 22.5%, soil loss rate decreased to 5.4%.

The biomass of *Sporosarcina pasteurii* is generally proportional to the amount of urease produced. After culturing, the bacteria surface with a strong negative charge is conducive to the formation of $CaCO_3$ through the combination of $Ca^{2+}$, which is highly adaptable to the environment [48]. In addition to the urea-producing bacteria, the presence of non-urease bacteria also accelerates the MICP reaction process. Gat et al. [47] studied the influence of the interaction between *Sporosarcina pasteurii* and non-urease bacterium *Bacillus subtilis* on the MICP effect, and showed that the rate of precipitation $CaCO_3$ from both bacteria was faster than that from *Sporosarcina pasteurii* alone.

In the case of low urease activity of the local bacteria, the introduction of exogenous urease bacteria to solidify the sand achieves good results. Tobler et al. [66] found that in the case of low urease activity of local bacteria, the introduction of *Sporosarcina pasteurii* promotes the rapid formation of precipitation in oxygenated and anoxic groundwater. Nevertheless, the cost of a urea-producing culture medium is high. In order to reduce the cost of culture medium for urea-producing bacteria, researchers have found economical alternative culture mediums [67]. The urease activity of the urease bacteria cultured by the alternative culture medium is almost the same as that of the urease bacteria cultured by the standard culture medium. Table 2 lists some economic mediums which have been found to replace standard media for urea-producing microorganisms. Figure 10 [68] shows the metabolic process of urease-producing bacteria using dairy wastewater.

**Table 2.** Economical alternatives to medium nutrient sources.

| The Name of the Bacteria | Economical Alternatives | Substitution of Nutrients in the Medium | References, Year |
|---|---|---|---|
| *Sporosarcina pasteurii* | Corn-steep liquor | Protein | [69], 2011 |
| *Sporosarcina pasteurii* NCIM 2477 | Lactose mother liquor | Protein | [70], 2009 |
| *Sporosarcina pasteurii* NB28 | Food-grade yeast | Nitrogen source | [71], 2019 |
| *Sporosarcina pasteurii* | By-products coming from the dairy and brewery industries | Protein | [72], 2015 |
| | Fertilizer urea | Urea | |
| *Sporosarcina pasteurii* isolated from agricultural soils of Sotaquirá and Nobsa | Whey | Protein | [73], 2021 |
| *S. pasteurii* DSMZ 33 | Corn-steep liquor | Carbon source | [74], 2018 |
| *Bacillus pasteurii* KCTC 3558 | Effluent from chicken manure bio-gas plant | Protein | [75], 2016 |
| *Psychrobacillus* sp. | Beer Yeast | Nitrogen source | [76], 2019 |

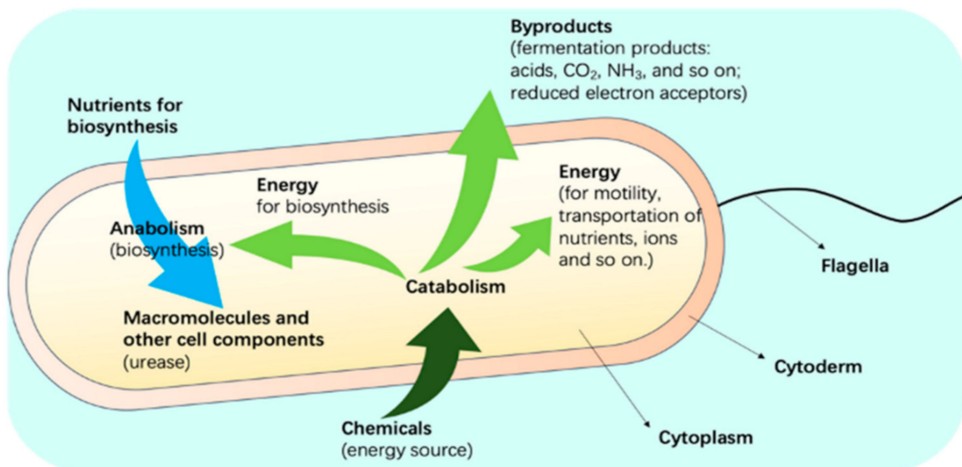

**Figure 10.** The metabolic process of urease-producing bacteria using dairy wastewater [68].

*2.2. Other Sand Fixation Methods*

At present, in addition to MICP sand fixation, the sand barrier, chemical sand consolidation, and solidification of sand by enzyme-induced carbonate precipitation (EICP) technology can also be used to fix sand.

Sand fixation with the sand barrier is a sand barrier made of clay, fences, and plants. By changing the properties of the underlying surface and increasing the roughness of the surface, sand barriers can prevent wind and sand consolidation and reduce the impact of wind and rain on sand erosion [77]. Chemical sand consolidation involves spraying chemical adhesive materials on the surface of sandy soil to make the surface sand particles bond to or infiltrate the chemical bonding material into sandy soil, so that the internal sand particles bond together to form a protective layer that plays the role of wind and sand consolidation, and improves the permeability of water in sandy soil. MICP technology and EICP technology both belong to biological cementation technology. MICP induces calcium carbonate precipitation by bacteria, and EICP generates calcium carbonate precipitation through hydrolysis of urea catalyzed by plant-derived urease enzymes. The performance of EICP may be affected by the source and activity of the enzyme, the calcium concentration, and the EICP treatment method [78].

Sand fixation with the sand barrier, chemical sand fixation, solidification of sand by EICP technology, and microbial sand fixation have distinct advantages and disadvantages. The most suitable sand fixation technology should be selected according to local environmental conditions. Table 3 lists the advantages and disadvantages of sand barriers, solidification of sand by EICP technology, chemical, and MICP. The cost of sand barriers and chemical sand fixation is relatively low, and the cost of MICP and EICP technologies is high. Sand fixation with sand barrier has advantages, including good sand consolidation, environmental friendliness, and convenient operation. However, sand barriers of grass grids need to be set carefully, considering factors such as wind direction and the sand's direction with wind movement; otherwise, they may not affect sand consolidation [79]. Most chemical sand fixing materials have significant economic benefits, but the chemicals used are often toxic to the environment, which is not environmentally friendly. Sand fixation with the sand barrier cannot improve the characteristics of desert sand. Chemical sand fixation, the solidification of sand by EICP and MICP technologies, can improve the characteristics of desert sand. Compared with MICP, EICP can be applied to finer sand due to the smaller particle size of the enzyme than microbe [80], and there is no need to cultivate bacteria in the EICP. Therefore, EICP is easier to operate. However, the commercial pure urease enzyme is expensive, and extraction techniques may involve additional processes and chemicals [78]. MICP cannot be used to treat soil with pores smaller than 0.5 μm due to the diameter of the microbe being larger than 0.5 μm and the complexity of

bacterial cultures used in MICP [80]. Because the urease of MICP is produced by bacteria, an appropriate environment needs to be maintained to generate enough urease with high activity [80].

**Table 3.** Comparison of advantages and disadvantages of various sand consolidation methods.

| Sand Consolidation Method | Figure | References | Advantages | Disadvantages |
|---|---|---|---|---|
| Sand fixation with sand barrier |  | [9] | Good curing effect, use of plants for environmental pollution, simple operation and cost is not high. | Cannot improve sandy soil after desertification, but can play a role in curbing desertification. |
| Chemical sand fixation |  | [7] | High economic benefits and good curing effects. | Complex operation and most chemical hardeners pollute the environment. |
| Solidification of sand by enzyme-induced carbonate precipitation (EICP) technology |  | [81] | Can treat fine soil; easier operation. | High cost; the precipitated calcium carbonate may not bind to soil particles due to the lack of nucleation sites. |
| MICP sand fixation |  | [8] | A good sand consolidation effect; well improves the properties of sandy soil; environmental friendly. | Complex operation and high cost. |

(**a**) Full contact flexible mold; (**b**) samples before the MICP reactions; (**c**) samples after the MICP reactions; and (**d**) sample after cutting mold.

## 3. Microbial Sources of Solidified Sand

There are two methods of microbially solidifying soil: one is the introduction of exogenous bacterially solidified soil, but as there are plenty of other microbes in the sand, the exogenous bacteria may compete with the existing microbes for nutrients. Therefore, it is necessary to add exogenous bacterial fluid constantly, which results in high cost. The other method is to use urea-producing bacteria existing in sandy soil, not by introducing exogenous bacteria, but by adding cementation solution and nutrients that facilitate the growth of indigenous bacteria. These two sand consolidation methods will affect the size and quantity of the calcium carbonate generated. Gomez et al. [82] found that compared with the calcium carbonate generated by exogenous bacteria, the size and quantity of calcium carbonate crystals generated by indigenous bacteria solidified sand soil were larger and fewer.

### 3.1. External Bacteria Solidifying the Sandy Soil

There have been many studies on introducing exogenous bacteria in order to solidify sandy soil. It is necessary to add cementing fluid to stimulate indigenous bacteria

to produce calcium carbonate precipitation, and at the same time, introduce exogenous urease-producing bacteria that can enhance the effect of microbial sand consolidation. Bernardi et al. [83] solidified sand with *Sporosarcina paseurii*, and the minimum porosity ratio of sand was 0.5. When the concentration of urea was 200 mM, the concentration of calcium chloride was 100 mM, and the $OD_{600}$ of the bacterial solution was one, the porosity ratio of sand samples after 28 days of treatment decreased to about 0.33, because the generated calcium carbonate was blocked in the original gaps in the sandy soil. Nafisi et al. [84] compared the effect of curing silica sand with *Sporosarcina pasteurii* and urease powder. They found that compared with curing silica sand with urease powder, curing silica sand with *Sporosarcina pasteurii* generated more calcium carbonate, and the shear strength of the solidified sand sample was greater. However, Ahenkorah et al. [85] compared the mechanical properties of sand samples solidified by MICP and EICP, and found that the splitting tensile strength of sand treated by EICP is higher than that of MICP. Cheng et al. [86] solidified the sand by a single-phase injection of low pH integrated solution into the sand, and mixed $OD_{600}$ of 4.2 *Bacillus* sp. with 1 M urea-calcium chloride solution to form an integrated solution. The pH of the solution was adjusted to four, and the rate of solution transmission was 1 L/h. After six times of treatments, the compressive strength of the sand sample reached 2.5 MPa.

Exogenous bacteria are also used in curing sand in the sea; Cheng et al. [49] proposed an innovative method of biological sand fixation method, which, without introducing $Ca^{2+}$, the $Ca^{2+}$ contained in seawater was used as the sole source of calcium, and then *Bacillus* sp. was introduced to solidify the sand in the seawater environment. The experimental results showed the feasibility of using $Ca^{2+}$ from seawater to solidify sandy soil, including that seawater can be used many times and is beneficial to improving the mechanical performance of sandy soil. The use of MICP technology for biocementation in the seawater environment provides a potential method for land reclamation. Xu et al. [87] proposed an experimental scheme similar to that of Cheng, using no additional introduced exogenous $Ca^{2+}$, and only using $Ca^{2+}$ from the fly ash of municipal incineration waste. The ratio of the fly ash to *S. pasteurii* bacterial solution was 1 kg:0.3 L. At 20 °C, humidity is not less than 95% for the 7 days curing experiment environment. The results show that the leaching rate of heavy metals decreases obviously after the solidification of fly ash, and the compressive strength increases by nearly 40% compared with that before the solidification. Wang et al. [88] used MICP technology to reduce the wind erosion rate of sandy soil. Their results showed that the wind erosion rate of untreated sandy soil was 10.23%, but when MICP treatment times were more than three times, the wind erosion rate of sandy soil dropped below 0.4%. Wind erosion rate is the ratio of the mass of the remaining sand that has been blown by the wind to the mass of the original sand that has not been blown.

Using exogenous bacteria to solidify sand requires the addition of bacterial liquid and cementation solution. The newly added exogenous bacteria will compete with the bacteria inside the sandy soil, so the bacterial solution needs to be added at intervals to ensure that the exogenous bacteria survive. Many studies have shown that exogenous bacteria can solidify soil, but by adding exogenous bacteria to the sand, it can be found that there is a lot of precipitation generated at the filling mouth, and the sediment distribution in the sand is not uniform. Moreover, the introduction of exogenous bacteria may not be conducive to the protection of ancient buildings, such as the reinforcement of the surface of ancient buildings and the repair of cracks, etc., the introduction of exogenous bacteria may destroy the dynamic balance of the bacterial community inside the original ancient buildings, and may cause secondary damage to the ancient buildings [89]. Therefore, solving these problems can be the direction of future research.

### 3.2. Solidification of Sandy Soil by Indigenous Bacteria

The indigenous bacteria themselves exist in the sandy soil and have strong adaptability to the environment. There are two ways to solidify sandy soil with indigenous bacteria. The first is to screen out indigenous bacteria from the soil for culture and then add the

bacterial culture solution and cementation solution in the sand; the other is to directly add nutrients for the in-situ culture of bacteria and then add the cementation solution to the sand. The utilization of indigenous bacteria is economical and effective, causes less environmental pollution, and may lead to the uniform distribution of induced $CaCO_3$ precipitation [90]. The introduction of indigenous bacteria can be used for the conservation of ancient buildings. It can be seen from Figure 11 that the untreated surface of ancient buildings in the $MgSO_4$ solution has a fast dissolving rate, and after dealing with the indigenous bacteria on the ancient building surface, the surface of the ancient building was not obviously dissolved under $MgSO_4$ solution erosion, indicating that indigenous bacteria carried out on the ancient surface treatment can reduce the salt chemical weathering and thus protect the surface of ancient buildings [89].

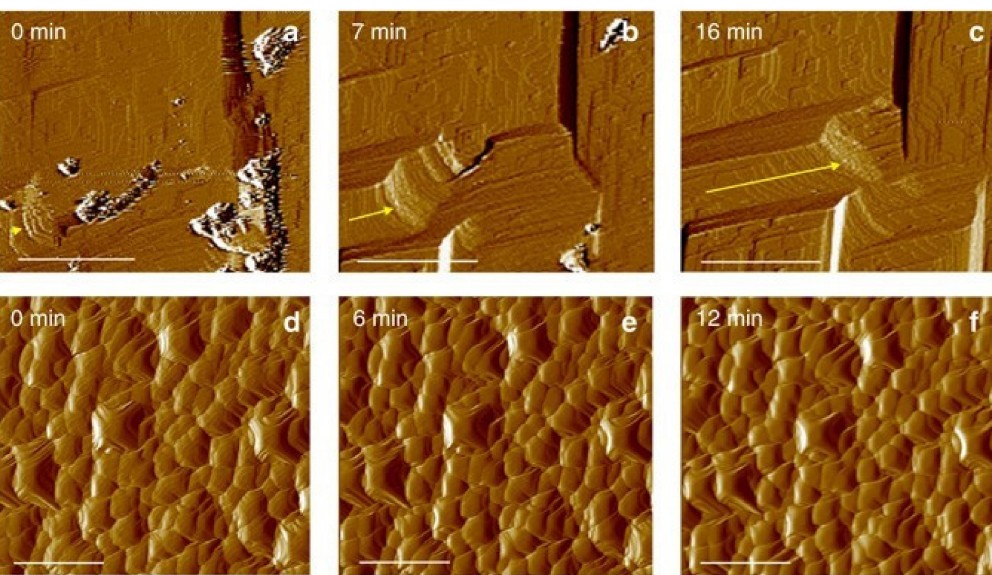

**Figure 11.** AFM image of the calcite surface over time after exposing the ancient building surface to MgSO4 solution. (**a**–**c**) Untreated; (**d**–**f**) Indigenous bacteria-treated [89].

Cheng et al. [91] enriched urease bacteria in soil and added the cementation solution to the soil for in-situ curing experiments. The results showed that in-situ curing does not cause surface blockage on the 1 m-high soil column, but the method of injecting the cementation solution should be constantly optimized in order to achieve deeper curing. Burbank et al. [92] found that urease-producing micro-organisms can be isolated in soils lacking urea or a high concentration of ammonia, and these microorganisms could be applied to mineralize the soil and repair the existing cracks in the soil. Kumari et al. [50] reported that MICP could fix Cd in the soil at low temperatures. *E. undae* YR10 isolated from the Yangtze River basin near Chongming Island fixed Cd in farmland soil near Chongming Island at 10 °C and 25 °C, and then converted Cd into components in carbonate. Burbank et al. [93] selected the soil with high urease content for microscopic tests and cyclic triaxial shear tests and found that not introducing exogenous bacteria and directly adding mineral solution can induce calcite precipitation, so as to improve the anti-liquefaction ability of sand; this method can achieve greater economic benefits than the addition of exogenous bacteria. Chahal et al. [94] screened and isolated urease-producing bacteria from alkaline soil to repair cracks formed in concrete and improve the life of the concrete. Gowthaman et al. [53] successfully isolated *Lysinibacillus xylanilyticus*, a bacterium from the sub-arctic region, which can produce urease at low temperatures, and successfully applied this bacterium to a slope-improvement project. The greater the urease activity of urease-producing bacteria, the better its effect in solidifying sand. Wang et al. [95] screened urea-producing bacteria from beach sand and studied the effects of different media and urea concentrations on bacterial urease activity. The experiment results showed that

nitrogen-rich composite media such as YE and NB increase bacterial urease activity, and urease activity at 100 mM concentration is highly efficient. Khan et al. [96] isolated a urease bacterium named *Parahodobacter* sp. from a beach of coral sand. *Parahodobacter* sp. was used to cure the coral sand for 28 days, and the compressive strength of the solidified coral sand reached 20 MPa. Oualha et al. [97] screened and isolated two strains of indigenous bacteria *B. cereus* from Qatari soil, namely QBB4 and QBB5. Both of these strains can solidify Qatari soil with a high pH and in a poor environment, and in a field experiment, the $CaCO_3$ content generated by soil solidified with QBB4 increased by 16.2% compared with the original soil. Song et al. [98] isolated *Staphylococcus succinus*J3 with high urease activity from the soil in a mining area and showed that the application of coal ash in a bacterially mineralized mining area could play a positive role. After solidification, the maximum wind speed of coal ash reached 45.5 m/s, and the maximum wind pressure reached 912 kPa. For the first time, Imran et al. [99] isolated indigenous urea-lytic bacteria from coastal erosion areas in Greece and showed that $CaCO_3$ could be generated, effectively protecting the coast from erosion. Chu et al. [100] isolated urease-producing bacterium (UPB) VS1 from tropical beaches and found that the solution of urea and calcium chloride added was lower than the sand surface, and calcium carbonate is evenly distributed in the sand. The solution of urea and calcium chloride added was higher than the sand surface, and the resulting calcium carbonate formed a solid shell on the sand surface.

## 4. Models for Predicting the Curing Process of MICP in the Field

The technology of microbial solidification of sand is relatively mature in the laboratory. For example, Phillips et al. [101] repaired sandstone fractures using two grouting methods and the experiment showed that multiple grouting methods promote the even distribution of $CaCO_3$ deposit in sandstone along the inflow direction. However, excessive repeated treatment leads to deposition blockage near the injection point [102]. Microbial solidification of sand has also been used in the field. Cuthbert et al. [103] applied MICP technology to fractured rocks and showed that when bacterial fluid and urea are simultaneously injected into fractured rocks, the addition of $CaCl_2$ solution promotes the formation of $CaCO_3$ precipitation to repair cracks, thus significantly reducing the permeability of rocks.

However, field tests need to overcome difficulties caused by many environmental factors and are very expensive [104]. Therefore, field tests are rare. Harkes et al. [105] used the two-stage method of adding bacterial solution and fixation solution in order to make the calcium carbonate generated by MICP evenly distributed in the sand, but the operation is complicated and the economic cost is high. Due to the complexity of the natural environment, some phenomena are difficult to explain. Ohan et al. [106] found that after applying MICP, the pH value of groundwater decreased, which contradicted the normal pH value increase.

The researchers found that models could be used to better analyze the dynamic changes and reaction mechanisms of the MICP process. A good model can predict the mechanical properties of MICP solidified sand samples and is helpful for the engineering design [107]. Table 4 lists some models that predict the curing process of MICP. Fauriel et al. [108] proposed a prediction model of the microbial grouting response based on the changes in porosity, permeability and density of soil after microbial grouting. Connolly et al. [109] introduced urease genes into *P. aeruginosa* AH298 and *E. coli* AF504gfp to construct two urease strains of *Pseudomonas aeruginosa* MJK1 and *Escherichia coli* MJK2 that had a characteristic of expressing green fluorescent protein (GFP), and used the Gompertz function to model the bacterial population density. It was found that the urealytic rate of the two strains was not high, *Escherichia coli* MJK2 grew faster, and *Pseudomonas aeruginosa* MJK1 had a higher urealytic rate. Gai et al. [110] established a model to evaluate the mechanical properties of MICP-solidified sandy soil, which clearly reflects changes in the mechanical properties of solidified sand soil, and the analysis of model parameters and the law of mechanical properties change are helpful to understand the process of MICP solidification. The results showed that mechanical properties are related to $CaCO_3$

content. Wang et al. [111] established a biochemical-hydraulic model, which proposes the concept of porosity to reflect the change of permeability. Their results showed that the pore structure has an important influence on the curing rate, the maximum urease rate has an indispensable influence on the hydraulic response of MICP, and the MICP reaction rate is influenced by the concentration of the bacterial and cementation solutions. Martinez et al. [112] proposed a biological reaction migration model, which coupled UCODE-2005 with TOUGHREACT sequence, and its practicability was confirmed in the MICP prediction experiment. TOUGHREACT numerical simulation program was used to reflect the reaction rate of urea hydrolysis and $CaCO_3$ generation in the MICP process. UCODE-2005 model was used to correct and verify the MICP experimental data. The results showed that the actual experimental results are close to the predicted data in the half-meter sand column experiment and dynamic changes in the MICP process can be seen.

**Table 4.** Models for predicting the curing process of MICP.

| Model Names | The Role of Models | References, Year |
|---|---|---|
| Aquifer conceptual model | Finding that the sedimentation rate of calcite is closely related to the hydrolysis rate of urea | [113], 2005 |
| A three-dimensional (3D) discrete element method (DEM)-based numerical model | Simulating the macroscopic mechanical properties of $CaCO_3$ sediment-solidified sandy soil induced by micro-organisms under the condition of no triaxial compression of the drainage system | [114], 2019 |
| A loose sandstone numerical model based on a one-dimensional advective dispersion model | Predicting the movement of micro-organisms in soil and rock | [115], 2014 |
| A pore-scale network model | Simulating the $CaCO_3$ precipitation process and the influence of different operations on $CaCO_3$ precipitation | [116], 2016 |
| Thermal conductivity predictive models | Predicting the thermal conductivities of MICP-treated sands | [117], 2020 |
| A small repeated five-point treatment model | Predicting solidification treatment in large-scale field experiments | [118], 2014 |
| The biogrouting foam model | Simulating key solidification processes such as on-site bacterial solution perfusion and adhesion and urea hydrolysis | [119], 2019 |

The solidification effect and mechanical properties of MICP can be simulated by the model. Before the large-scale outdoor experiment, model analysis of the existing laboratory-scale experimental data can be carried out to predict the results of the large-scale outdoor experiment. The combination of model analysis and laboratory data is conducive to the smooth implementation of large-scale field experiments [27,120]. At present, researchers have established many models, and each model has its own role. In the future, multiple models can be combined to improve the accuracy of the model prediction.

## 5. Factors Affecting Microbial Solidification of Sandy Soil

The effect of microbial solidification of sandy soil is influenced by a variety of factors, including the concentration of the cementation solution, the concentration of culture liquid, temperature, calcium source and pH value. Before large-scale experiments, the

effects of different factors on the solidification effect should be studied to optimize the experimental conditions and determine the optimal solidification conditions. Under the same conditions, the curing effect of the same bacteria on different sandy soil is different. Table 5 shows the optimal conditions for microbial solidification determined by researchers when conducting MICP.

**Table 5.** Results on the optimal curing conditions of urease-producing bacteria.

| Microbe Name | Optimum Curing Condition | | | | Reference, Year |
|---|---|---|---|---|---|
| | Microbial Concentration | The Concentration of Cementation Solution | Temperature | Cure Time | |
| *Sporosarcina pasteurii* | $1 \times 10^7$ cells/mL | 0.5 M urea-CaCl$_2$ | 20 °C | 16 days | [121], 2012 |
| *B. megaterium* | $1 \times 10^8$ cfu/mL | 0.5 M urea-CaCl$_2$ | 22–27 °C | 48 h | [122], 2014 |
| *Parahodobacter* sp. | $10^9$ cfu/mL | 0.5 M urea-CaCl$_2$ | 30 °C | 21 days | [96], 2015 |
| *Sporosarcina pasteurii* | OD$_{600}$ = 4 | 3.0 M urea, 1.5 M CaCl$_2$ | 30 °C | 21 days | [123], 2016 |
| *Staphylococcus succinu* | OD$_{600}$ = 0.7 | 40 mmol Ca$^{2+}$, 6%(*w/w*) urea | 30 °C | 35 days | [98], 2019 |
| *S. aquimarina* | $12.8 \times 10^9$ cells/ml | 0·25 M urea, 2 M CaCl$_2$ | — | — | [124], 2018 |
| *S. pasteurii* | $9 \times 10^9$ cells/ml | 1 M urea, 2 M CaCl$_2$ | — | — | |
| *Pararhodobacter* sp. | — | 0.5 M CaCl$_2$ | 25 °C | 14 days | [44], 2018 |
| *Sporosarcina pasteurii* | $1 \times 10^8$ cells/mL | 50 mM Ca$^{2+}$ | — | — | [125], 2020 |

### 5.1. Concentration of Cementation Solution

To induce CaCO$_3$ precipitation with micro-organisms, cementation solution needs to be added. Cementation solution refers to the calcium source and urea solution. Different concentrations of cementation solution affect the compressive strength and permeability of sandy soil after cementation. In addition, the size and distribution of calcium carbonate generated by different concentrations of cementation solution in sandy soil are also different. Al Qabany et al. [121] studied the influence of different cementing fluid concentrations on solidified sand and found that the size of calcium carbonate particles generated after the treatment of 0.25 M urea-CaCl$_2$ was similar and distributed evenly, the size of calcium carbonate particles generated after the treatment of 0.5 M urea-CaCl$_2$ was different and distributed unevenly, and the size of calcium carbonate generated after treatment with 1 M urea-CaCl$_2$ was large and distributed unevenly. In a word, as the concentration of cementation solution increased, the surface strength of the solidified sample increased and more CaCO$_3$ content was precipitated. However, there is no correlation between surface strength and CaCO$_3$ content [126]. Too low a concentration of Ca$^{2+}$ is not conducive to the formation of CaCO$_3$, while too high a concentration of Ca$^{2+}$ may inhibit the urease activity of micro-organisms and affect the formation of CaCO$_3$ precipitation [127]. The concentration of urea can also affect the effect of microorganism sand fixation; properly increasing the concentration of urea will increase the production of calcium carbonate [128]. Li et al. [129] found that as the concentration of the solution of solidified aeolian sand increased, the CaCO$_3$ content increased, leading to increased sand density. Qabany et al. [130] used a 0.1–1 mol/L urea-CaCl$_2$ solution and $10^7$ cells/mL *Sporosarcina pasteurii* bacterial solution to cement sandy soil, and found that sandy soil treated with a high concentration of cementation solution had lower permeability than that treated with a low concentration of cementation solution, but the low concentration of cementation solution which was repeatedly fed into the resulting sediment was more uniform. Ng Wei Soon et al. [122] used the solution of *B. megaterium* and solution of urea-calcium chloride to solidify residual soil with a maximum dry density of 1688.5 kg/m$^3$, and found that when the concentration of urea and calcium chloride were both 0.5 mol/L, the concentration of B. megaterium was $1 \times 10^8$ cfu/mL, and the flow pressure of the cementation solution was 1.1 bar. The treatment time was 48 h, the engineering properties of the residual soil were improved the most; the shear strength of the cured sand was increased by 69%. Their microbial solidification experiment was carried out on the interaction between the cementation solution and the bacterial solution, and the concentrations of the two affected and restricted each other.

Mujah et al. [131] used the solution of *Bacillus* sp. and solution of urea-calcium chloride dehydrate to solidify silica sand with a maximum dry density of 16.3 kN/m$^3$, and found that the optimal solidification scheme was 32 U/mL urease concentration of *Bacillus* sp. bacterial solution and 0.25 M concentration of urea-calcium chloride dehydrate, and this could form rhombic calcite of a relatively large size. This crystal type of CaCO$_3$ can not only significantly improve the strength and stiffness of soil, but also maintain the permeability of soil samples. Whitaker et al. [132] found that *S. ureae* had a stronger curing capacity than *S. pasteurii*, which has been widely studied. When the researchers used *S. ureae* bacteria solution and added 0.5 M cementation solution, the direct shear strength of the solidified sandy soil increased from 15.77 kPa before solidification to 135.80 kPa. Chen et al. [133] used a strain of *bacillus* isolated from a mine tailing soil to solidify sandy soil, and found that when a cementation solution consisting of 15 mM Ca$^{2+}$ and 20 g/L urea was added to bacterial liquid with an OD$_{600}$ value of 0.4, the solidification effect was the best and the cost was the lowest. Jiang et al. [134] used $1.91 \times 10^8$ cells/mL of *Sporosarcina pasteurii* to improve the erosion resistance of silica sand with particle sizes ranging from 100 to 500 μm, and showed that the addition of 0.2 M and 1.0 M concentrations of urea-CaCl$_2$ solution improved the erosion resistance of the slope, while the addition of 2.0 M concentration of urea-CaCl$_2$ solution did not improve the erosion resistance of the slope.

In summary, the concentration of cementation solution will affect the size and content of calcium carbonate and its distribution uniformity in the sand. Most studies optimized the concentration of urea solution and calcium ion as the overall concentration of cementation solution, but calcium ion concentration and urea concentration both affect the effect of microbial sand consolidation, so the concentration of urea and calcium ions in cementation solution can be optimized, respectively, in the future.

### 5.2. Bacterial Concentration

In the process of the introduction of exogenous bacteria to solidify soil, a different concentration of bacterial solution has a different effect on the curing effect. The determination of the optimum microbial concentration is beneficial to improve the performance of cured samples. The optimum concentration of the same bacteria needed to irrigate sand from different sources was different, and the optimum concentration of bacteria solidified by the same source sand was also different.

*B. pasteurii* is usually used as an exobacterium, and Chahal et al. [135] determined through experiments that the optimal concentration of *B. pasteurii* bacterial solution is 10$^5$ cells/mL. After 91 days of curing, the compressive strength of concrete was significantly improved compared with that of untreated concrete, with an increase of 44 MPa. The increase in compressive strength of the solidified samples was due to the increase in calcium carbonate precipitation induced by bacteria. The concentration of the bacterial solution will affect the rate of calcium carbonate precipitation. Wen et al. [136] used *Sporosarcina pasteurii* and urease to cure sandy soil. Figure 12 [136] shows SEM images of CaCO$_3$ after treatment with different concentrations of bacterial liquid. As can be seen from Figure 12, when the OD600 of the bacterial solution was 0.1, flower-like crystals were precipitated at 168 h, while when the OD600 of the bacterial solution exceeded 0.1, flower-like crystals were precipitated at 72 h. Andalib et al. [137] found that the compressive strength and flexural strength of a bacterial concrete sample made of $30 \times 10^5$ cfu/mL *Bacillus megaterium* were the largest. When the bacterial concentration exceeded $30 \times 10^5$ cfu/mL, the compressive strength and flexural strength of the sample decreased.

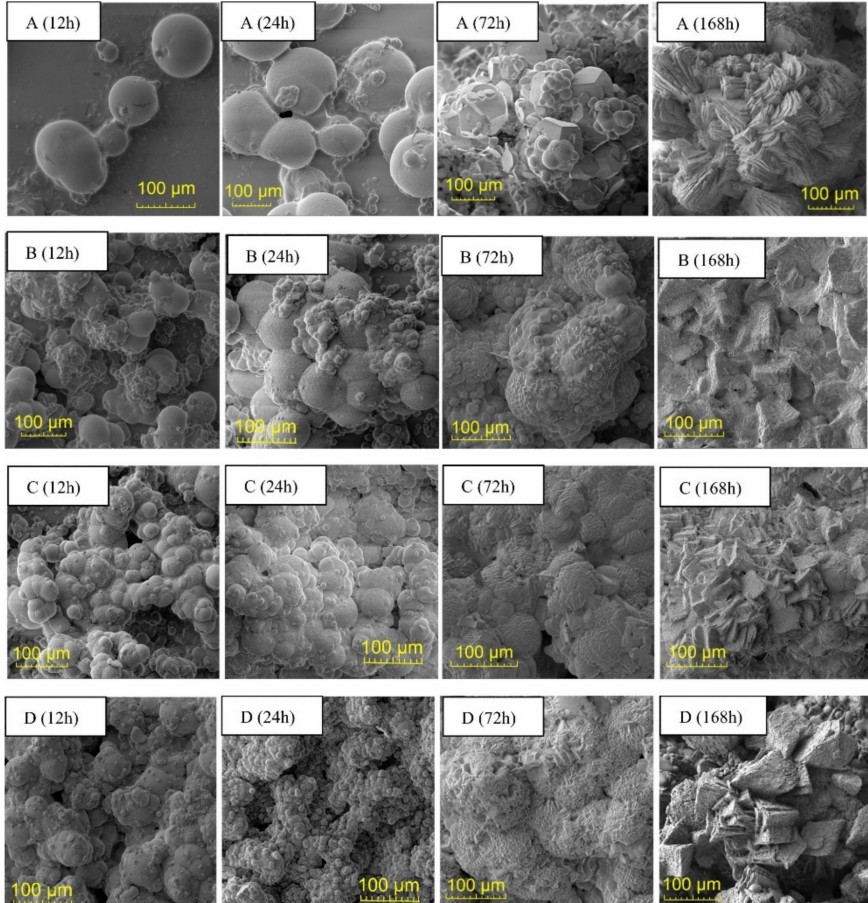

**Figure 12.** SEM images of $CaCO_3$ treated with different concentrations of bacterial solution. (**A**) $OD_{600} = 0.1$; (**B**) $OD_{600} = 0.3$; (**C**) $OD_{600} = 0.6$; (**D**) $OD_{600} = 1.0$ [136].

In summary, the concentration of the bacterial liquid will affect the sedimentation rate and crystal formation of calcium carbonate [136]. The concentration of bacteria solution is an important factor affecting the effect of microbial sand consolidation. Optimizing the concentration of bacteria solution is beneficial to improving the mechanical properties of sand. In the future, the economical medium composition that can achieve the optimal concentration of bacteria solution can be found to reduce the cost when achieving the same sand fixation effect.

### 5.3. Temperature

The temperature will affect the urease activity of urease bacteria, bacteria growth ability, and the production of calcium carbonate. Therefore, temperature control is very important in the application of microbial sand fixation. The calcite precipitated at different temperatures has different shapes [31]. The optimal temperature for urease-catalyzed hydrolysis of urea is generally between 20 °C and 37 °C. Figure 13 [138] shows the effect of temperature on the urease activity of some urease-producing bacteria. It can be seen from Figure 13 that the urease activity of these five strains was the highest when cultured at 25–30 °C. However, Fujita et al. [139] isolated *Pararhodobacter* sp. SO1 from nearby Okinawa, Japan. This bacterium showed the highest urease activity under 60 °C environment, and the urease activity decreased over 60 °C. The optimal temperature conditions for the formation of $CaCO_3$ from urea catalyzed by different urease-producing bacteria are also different [28]. Kim et al. [140] found that the optimal induced calcite precipitation temperature of urease-producing bacteria *B. pasteurii* was 30 °C, and Imran et al. [141] found that for urease-producing bacteria *Pararhodobacter* sp., the optimal $CaCO_3$ generation temperature was 35 °C. Deng et al. [142] found that when the temperature of *Sporosarcina*

*pasteurii* was 10 °C, the bacterial $OD_{600}$ was less than 1.0, which reduces the effect of the solidified sand. The optimal temperature for *Sporosarcina pasteurii* growth was 30–35 °C. Cheng et al. [143] simulated the solidification effect in cold, tropical and arid regions, and selected three temperatures, 4 °C, 25 °C and 50 °C, for their experiment. The results showed that compressive strength after curing increases with the increase in the precipitated $CaCO_3$ content at any temperature. The optimal curing temperature is 25 °C. The size of a single calcite crystal cured at 25 °C is about 10 times that of a single calcite crystal cured at 4 °C or 50 °C. Large-size calcium carbonate crystals can effectively fill the gap of the sand, which is beneficial to increase the strength of the solidified sand.

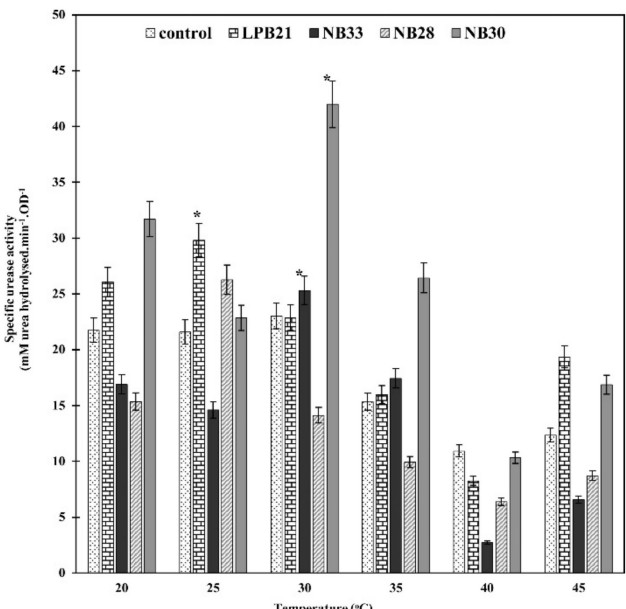

**Figure 13.** Effect of temperature on urease activity of urease-producing bacteria. Control: *S. pasteurii* DSM 33; LPB21: closest match *Sporosarcina pasteurii* fwzy14; NB33: closest match *Sporosarcina pasteurii* WJ-4; NB28: closest match *Sporosarcina pasteurii* WJ-5; NB30: closest match *Sporosarcina pasteurii* fwzy14. Significance level was set to 0.05 (*) [138].

Temperature affects the permeability of the oil reservoir by urease solidification, but the solidification effect of temperature on oil reservoirs does not have a single cause, and is also affected by urease concentration. With the increase in temperature, the consolidation capacity of a high-concentration urease solution is significantly improved and the permeability is constantly enhanced, while a low-concentration urease solution shows little change in permeability with the increase in temperature [17]. In places with too high or too low temperatures, the growth of some micro-organisms is inhibited, posing a challenge for sand consolidation. Suitable curing sites should be selected according to the growth characteristics of the micro-organisms.

In summary, the culture temperature of bacteria will affect the growth ability and urease activity of bacteria, and the urease activity of bacteria will affect the speed of urea decomposition by bacteria, thus leading to the rate of calcium carbonate precipitation MICP reaction temperature influencing the crystal shape and size of calcium carbonate. Therefore, culture temperature and MICP reaction temperature are important factors affecting the MICP effect. At present, there are few studies on the effect of MICP reaction temperature on microbial sand fixation. Next, the influence of different reaction temperatures on microbial sand fixation should be focused on.

*5.4. pH Value*

Urease-producing bacteria hydrolyze urea and increase the pH value, and when the pH reaches a certain value, the urease activity is higher, which is conducive to the

generation of CaCO₃ in the MICP process. The initial pH of the medium can affect the urease activity of the urease-producing bacteria. Figure 14 [138] and Figure 15 [144] showed the effect of the initial medium pH on the urease activity of some urease-producing bacteria. The results in Figure 14 showed that the optimal initial medium pH of these urea-producing bacteria was 6.5–8, and the results in Figure 15 showed that the optimal initial medium pH of the isolated urea-producing bacteria was 10. Wu et al. [145] found that when the pH of the culture medium was eight, the urease activity of *B. cereus* CS1 reached the highest value. Alonso et al. [146] found that the urease activity of the screened ureolytic bacterial strains reached the highest value when the pH was eight. Microbial solidification requires the addition of bacterial solution and cementation solution to sandy soil. Kim et al. [140] found that the optimal solidification condition for *Staphylococcus saprophyticus* and *Sporosarcina pasteurii* was when the pH of the cementation solution was seven.

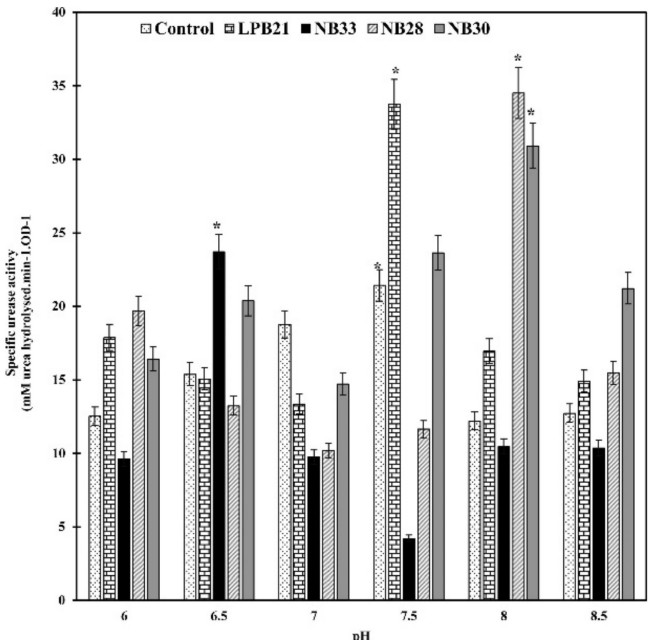

**Figure 14.** Effect of different initial medium pH on urease activity. Control: *S. pasteurii* DSM 33; LPB21: closest match *Sporosarcina pasteurii* fwzy14; NB33: closest match *Sporosarcina pasteurii* WJ-4; NB28: closest match *Sporosarcina pasteurii* WJ-5; NB30: closest match *Sporosarcina pasteurii* fwzy14. Significance level was set to 0.05 (*) [138].

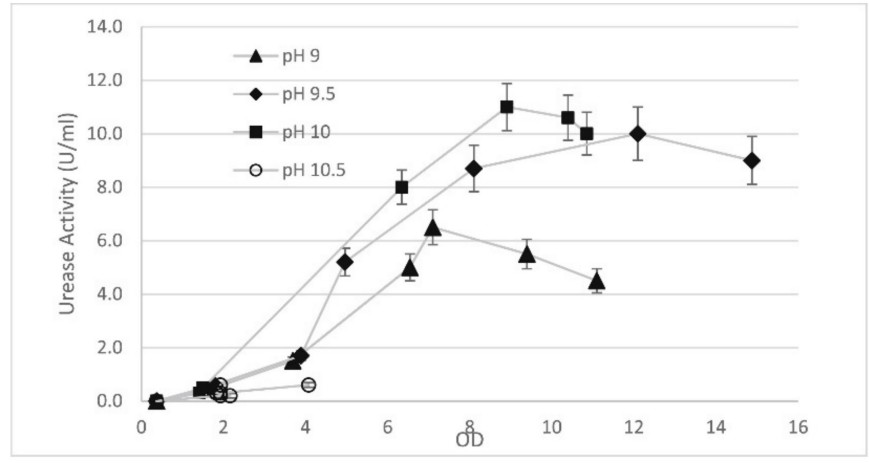

**Figure 15.** Relationship between urease activity and bacterial concentration under different initial pH mediums [144].

The pH value has an effect on the activity of functional groups, metal ions, and hydride binding sites, which is an important factor affecting the ability of MICP to remove metal ions from sandy soil. Zhao et al. [147] isolated urease bacteria GZ-22 from mining soil and used this bacterium to remove $Cd^{2+}$. When the pH of cadmium ion solution was six, the bacteria could precipitate the most $Cd^{2+}$ precipitation, and the removal efficiency of $Cd^{2+}$ reached 50.34%.

pH has the function of regulating the flocculation state of bacterial liquid and cementation solution. Wu et al. [148] prepared the mixed solution, which is composed of bacterial liquid and cementation solution. The mixed solution not adjusted by pH is flocculated, the flocculation state disappears when pH is adjusted to 4–5. The results showed that the flocculation solution without pH regulation could only effectively treat sand with a particle size of more than 2 mm, while the non-flocculating solution adjusted by pH can effectively treat sand with any particle size.

In summary, in the growth environment, the pH value and urease activity of urease-producing bacteria will gradually increase. Most of the urease bacteria reported in the literature reached the best urease activity at pH 8. However, when exogenous bacteria are introduced into the sand, the pH of exogenous bacteria will decrease, which is not conducive to MICP. The MICP effect is also affected by the pH value of cementation solution. Therefore, the pH value of bacteria fluid and cementation solution is an important factor affecting the sand consolidation effect. According to the grouting method, the pH value of the bacteria liquid and the cementation solution can be independently optimized or the overall pH value of the bacteria liquid and the cementation solution can be optimized. If the step-by-step grouting method of the bacteria liquid and the cementation solution is adopted, the pH value of the bacteria liquid and the cementation solution can be independently optimized. If the one-step grouting method is adopted, the overall pH value of the bacteria liquid and the cementation solution can be optimized.

### 5.5. Sources of Calcium

Many calcium sources are used in the cementation solution to induce microorganisms to produce precipitation, including calcium chloride, calcium acetate, calcium gluconate and calcium lactate [149]. In recent years, researchers have proposed substitutes for commonly-used calcium salts, such as eggshells, seawater, papermaking wastewater, which are more economical and environmentally friendly [150]. Røyne et al. [151] proposed the use of limestone powder as the calcium source for MICP application. First, the limestone powder was dissolved by bacteria AP-004 screened and analyzed from soil near the quarry, and then urease-catalyzed urea hydrolysis was induced by *Sporosarcina pasteurii* to develop an adhesive, providing a new idea for the source of calcium salt. Choi et al. [152] dissolved limestone powder in acetic acid solution to form calcium ion solution, and found that when using *Sporosarcina pasteurii* solution with $OD_{600}$ of 0.8–1.2 and 0.3 M urea-$Ca^{2+}$ solution to cure sand column, the content of calcium carbonate generated was 8.19%. Choi et al. [153] used calcium salt produced by mixing eggshells and vinegar in a ratio of 1:8 as the calcium source for biosolidification. The results showed that there was little difference in the effect of sand solidification with calcium salt produced by mixing eggshell and vinegar, and $CaCl_2$ as the calcium source. Liang et al. [154] found that garbage in the kitchen, such as scallop shells, eggshells, and oyster shells, could be used as calcium sources for MICP and play a better role in solidifying sandy soil. Of these, the compressive strength of sandy soil solidified with oyster shells as the calcium source can reach 1454.6 KPa. Liu et al. [155] mixed acetic acid with calcium-containing sandy soil to dissolve the calcium in the sandy soil and compared the calcium dissolved in the sandy soil with a sand column treated with $CaCl_2$. The dry density and compressive strength of the sand column treated with calcium dissolved in sandy soil were higher than those treated with $CaCl_2$, reflecting the feasibility of using calcium dissolved in calcium-containing sand as the calcium source for MICP.

Different researchers have studied the influence of calcium salts on MICP technology and obtained different research results. Pan et al. [156] compared the effects of calcium

salts $CaCl_2$, $Ca(NO_3)_2$, and $Ca(CH_3COO)_2$ on *Bacillus cereus* solidified sand. The results showed that the $CaCO_3$ content precipitated from $CaCl_2$ and $Ca(CH_3COO)_2$ as calcium sources were relatively high, while $Ca(NO_3)_2$ as a calcium source produced more dense and uniform precipitation and greater compressive strength. Zhang et al. [157] compared sandy soil treated with $CaCl_2$, $Ca(CH_3COO)_2$, and $Ca(NO_3)_2$ as calcium sources and found that the uniaxial compressive strength of mortar treated with $Ca(CH_3COO)_2$ was more than twice that of the other two treatments, and the spatial distribution of mortar treated with $Ca(CH_3COO)_2$ was more uniform. Abo-El-Enein et al. [158] used *S. pasteurii* to solidify sandy soil, and 1 M $CaCl_2$, $Ca(CH_3COO)_2$, $Ca(NO_3)_2$, and 1 M urea was mixed in the cementation solution, respectively. The test results showed that the samples treated with $CaCl_2$ had more $CaCO_3$ production, higher compressive strength, and lower water absorption than those treated with the other two calcium salts. SEM images showed that the precipitates after solidification of different calcium salts also had different appearances.

To summarize, different calcium salts have different curing effects on sandy soil, so suitable calcium salts should be selected for specific sandy soil. In order to be more economical and environment-friendly, many researchers are actively looking for calcium sources that can replace calcium salts, such as eggshells and seawater. Calcium source is an important component of cementation solution, so finding the best calcium source is beneficial to improving the mechanical properties of solidified sand samples. In the future, more economical calcium salt substitutes can be found, and then the concentration of calcium salt substitutes can be further optimized.

## 6. Conclusions and Suggestions for Future Research

The problem of desertification is becoming more and more serious worldwide. In recent years, microbial sand consolidation technology has become a research hotspot in the field of sand consolidation due to its advantages of solidifying sandy soil, curbing desertification, increasing the permeability of sandy soil, and turning sandy soil into arable land.

The technique of calcium carbonate precipitation induced by micro-organisms has broad application prospects for the solidification of sand and the protection of building materials. The micro-organisms used to solidify sandy soil are derived from urease bacteria in sandy soil or the external environment. Urease-producing bacteria in sandy soil are generally more evenly distributed than the $CaCO_3$ produced by external bacteria. Numerical model analysis of existing laboratory data to simulate the parameters and expected results of large-scale experiments in the field can save costs and contribute to the smooth operation of large-scale experiments.

Many factors affect the microbial solidification of sandy soil, including the cementation solution concentration, bacterial liquid concentration, temperature, pH, and calcium salts. The concentration of cementation solution affects the size and distribution of calcium carbonate. In general, the concentration of the cementation solution is positively correlated with the curing effect. The content of calcium carbonate increases with the increase in the concentration of cementation solution. The deposition rate and crystal morphology of calcium carbonate were affected by the concentration of the bacterial solution. Therefore, it is necessary to determine the best concentration of bacteria to obtain the best curing effect. The culture temperature of the strain will affect the urease activity of the strain. The optimum culture temperature of different urease-producing bacteria was different, and the urease activity of most urease-producing bacteria reached the highest value when the culture temperature was 20–37 °C. The reaction temperature of MICP will affect the size of calcium carbonate. Many economic alternatives to calcium salts have been found, such as eggshells and seawater, etc. Calcium salts affect the precipitated $CaCO_3$ content and calcite crystal type. The pH of the bacterial solution affects urease activity, thus affecting the precipitation of calcite and removing heavy metals. The pH of the cementation solution also affects the amount of precipitate. Most urease-producing bacteria have the highest urease activity at pH 8.

Based on the analysis of existing experiments on microbial sand consolidation, some problems and suggested topics for future research are summarized as follows:

(1) At present, micro-organism solidification faces some problems. For example, when microbe grouting enters sand and other media, $CaCO_3$ deposit is not uniformly distributed, and induced $CaCO_3$ deposit is deposited near the injection port. This may be due to an imbalance between the rate of hydrolysis of urea and the rate of transport of the cementation solution to the sand or the rapid precipitation rate of calcium carbonate generated at the injection port [159,160]. It needs to add the cementation solution several times in batches, and the cost is relatively high. In addition, the sand body after grouting needs to be heated and dried, and the operation is complicated. Wang et al. [161] added anion admixture based on the concept of ionic lattice energy. The introduction of anion admixtures can combine $Ca^{2+}$ in the cementation solution to form ionic crystals, which can effectively reduce the number of grout times and avoid heating and drying this step after grout. However, the strength of sand consolidation is lower than that of other studies, so more efficient admixtures should be explored next.

(2) Urease-producing bacteria will produce the by-product of $NH_4^+$ in the process of catalyzing urea hydrolysis. A high concentration of $NH_4^+$ will harm the environment. At present, the research on ammonia removal mainly includes using natural zeolite to remove ammonium ions, the use of struvite to reduce ammonium by two steps, and reducing the pH in the reaction process to reduce ammonium ions [57–59]. More research on ammonium ion removal methods is needed in the future.

(3) There are also problems in solidifying sandy soil using the urease bacteria contained in the soil. It is troublesome to screen urease-producing bacteria in sandy soil, because of the need to carry out primary screening, re-screening, strain identification, and the urease activity of the selected strain may not be sufficient. However, the introduction of external bacteria could compete with bacteria inside the sand. In future, if measures can be taken to allow internal urea-producing bacteria and exogenous bacteria at the same time to solidify sand, this may be a promising research direction for the optimization of microbial sand consolidation.

(4) Although there are successful cases of microbial sand consolidation technology in field practice, there have been few large-scale experiments. Up until now, very few experiments on microbial sand consolidation technology have been conducted in deserts. Meng et al. [26] conducted field experiments with MICP in the Ulan Buh Desert and found that MICP can improve the resistance of sand to wind erosion. The high temperature of the desert made it easy for the solution to evaporate and posed a significant challenge to the survival of the species. Therefore, the treatment of sand by MICP is conducted after sunset. The effect of field sand consolidation experiments can be measured by the calcium carbonate content, the bearing capacity, and erosion depth of sand under wind conditions. When determining the optimal curing experimental conditions in the field, field environment and operation feasibility should be considered. The influences of temperature, pH, bacterial liquid concentration, cementation solution concentration, and calcium source on MICP should be studied. MICP-treated sand has poor durability in harsh environments, such as wet–dry cycles, freeze–thaw cycles, and acid rain conditions [8]. There are few studies on the durability of MICP-treated sand under acid rain conditions, which can be studied more in the future. In addition, future research can overcome the adverse conditions of high temperatures by mutating genetic strains, thus achieving large-scale sand consolidation in deserts.

(5) Sand fixation methods include sand fixation with the sand barrier, chemical sand consolidation, and microbial sand consolidation. Each method has its advantages and disadvantages. Currently, most researchers use one curing method to solidify sandy soil, but no researchers to date have combined multiple curing methods to

solidify sandy soil. This can be regarded as a future research direction in the field of solidifying sandy soil.

(6) Some experimental results have shown that the combined actions of non-urease bacteria and urease-producing bacteria enhance the microbial solidification of sandy soil. However, there have been few studies of the simultaneous use of non-urease bacteria and urease-producing bacteria to date. Gat et al. [162] found that the precipitation rate of calcium carbonate generated by non-urease bacteria *B. subtilis* and *S. pasteurii* was faster than that generated by *S. pasteurii* alone. They speculated that non-urease bacteria provided additional nucleation sites for MICP, which promoted the precipitation of calcium carbonate and accelerated the MICP process. This should be considered as a factor in the subsequent optimization of solidification experiments.

(7) The accuracy of the prediction results of the model will be affected by some factors. Kim et al. [163] found that the results predicted by the numerical model deviated from the actual value because the predicted results of the model were affected by factors such as the assumed shape of calcium carbonate particles and local pore blockage. Therefore, it was difficult to directly and accurately predict the pore-scale characteristics of MICP solidified sand through the model. Existing numerical models should be continuously optimized, and new numerical models should be proposed.

(8) Biological consolidation techniques mainly include the MICP method and another promising method, enzyme-induced carbonate precipitation (EICP) [164,165]. MICP produces urease using urease-producing bacteria in the culture environment, but the cultural environment of urease-producing bacteria is complicated, and urease activity is difficult to control [80]. EICP is derived from the direct use of free urease, and the enzymes can be derived from microbes, fungi, and agricultural sources. Microbially induced carbonate precipitation cannot treat sandy soils with pores smaller than 0.5 μm because bacteria are 0.5 to 3 μm in size. EICP can treat finer clays because the enzyme particle size is about 12 nm [80]. Compared with MICP, EICP has the advantage of not involving biosafety issues, while the calcium carbonate production efficiency of EICP is lower than that of MICP [166].

**Author Contributions:** L.C.: drafted the manuscript after conducting a literature search; Y.S.: initial idea, acquisition of funding, paper review & revision; X.S.: acquisition of funding, supervision, paper review & revision; H.F.: review & revision of manuscript; J.H.: paper review & revision; C.L.: paper review & revision; H.J.: paper review & revision; J.Z.: paper review & revision. All authors have read and agreed to the published version of the manuscript.

**Funding:** The authors gratefully acknowledge Nanjing Forestry University Key Laboratory of Forest Genetics and Biotechnology. The work was supported by the Natural Science Foundation of Jiangsu Province For Youth (BK20150874), and National Natural Science Foundation For Youth (31600463). The authors thank the Priority Academic Program Development of Jiangsu Higher Education Institution (PAPD) for supporting the work. This research is supported by the China Scholarship Council-Monash University Faculty of Engineering International Postgraduate Research Scholarship (CSC-Monash FEIPRS) joint project [Grant number: 201708320284].

**Informed Consent Statement:** Not applicable.

**Data Availability Statement:** Not applicable.

**Conflicts of Interest:** The authors declare no conflict of interest.

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
