# Peer review of "Critical Review of Solidification of Sandy Soil by Microbially Induced Carbonate Precipitation (MICP)"

_crystals, doi:10.3390/cryst11121439_

Round 1

Reviewer 1 Report

It nice paper to discuss. I believe that this paper brings a significant effect on the MICP research. 

I have several basic questions for this paper
1. Line 51-55: Authors need to explain what is the consolidated method and please add several reason and references related to comparison the grouting method and consolidated method

2. Figure 2. Please add these chematic for consolidated method too

3. Figure 3. It high recommends adding the schematic of the test

4. Conclusion section - It nice conclusion. If possible the author is recommended to add the challenge related to field test for MICP. Compare it to the other Calcite Precipitation Method, such as using the enzyme or the plant-derived urease

Reviewer 2 Report

The authors present a comprehensive review article on MICP based solidification of sandy soil. The review nicely presents the state-of the art based on the work performed so far, indicating some research gaps for future prospects. However, the following comments should be carefully addressed for publishing the article in "Crystals".

  1. First of all, authors should avoid using "I", "you", "we" in the article. Please revise the sentences appropriately.
  2. I wonder why some of the urease producing bacteria used in MICP are not listed in the Table 2. I recommend the authors to add the following bacteria to the table, Lysinibacillus xylanilyticus and Psychrobacillus sp (sourced from Hokkaido, Japan). This would make the review article more informative to the readers.
  3. Refer the line 186, refer Table 5: the "3" in CaCO3 should be subscript.
  4. The authors should be very careful when writing the scientific names of the bacterial species. For example, for the "Bacillus sp", the "Bacillus" should be in italic, but "sp." should not be in italic. The correct scientific way of writing is "Bacillus sp.". Please revise the names correctly throughout the manuscript.
  5. Some researchers have recently found that the "Beer Yeast" is one of the effective and economic alternative nutrient medium. It can be added to the Table 3 (authors may refer the following article, https://doi.org/10.1016/j.sandf.2018.12.010)
  6. In the abstract, it has been stated that "minimizing NH4+ production during MICP". However, there are no critical discussions in the text. I fact, the minimizing of ammonium by products is a crucial concern, thus recommending to be added in the article. Please refer more recent works that focus on ammonium removal and discuss in the manuscript to enhance the quality of the review work. For example, there are studies focusing on the use of zeolite material to adsorb ammonium produced in MICP, two-stage treatment to manage ammonium by-products in MICP by struvite formation, low pH MICP for minimizing the formation of ammonium gas, etc.
  7. The discussion on durability of MICP treated soil against various environmental factors are still lacking. Authors should refer more recent works and enhance the discussion, as this remains a great challenge to be addressed in the future works. For example, discuss the durability of treated soils under freeze-thaw cycles, wet-dry actions, etc. Acid rain fall is also another challenge to MICP treatment. Recently, there are few works addressed the durability of the MICP treatment under acid rain conditions. Authors may refer those and add useful information in this review.
  8. Correct the reference 117.

Round 2

Reviewer 2 Report

The authors have referred the recent research publications, enhanced the review and appropriately revised the manuscript. I recommend the article to be published in Crystals.

Author Response

The authors would like to thank the reviewer for your careful and patient reviews of the manuscript.  The suggestions made by the reviewer were salient for the communication of issues addressed in this paper. The authors feel that the comments from reviewer have greatly benefited the manuscript and are grateful for your time and effort.